# First-Principles Design of Refractory High Entropy Alloy VMoNbTaW

**DOI:** 10.3390/e20120965

**Published:** 2018-12-13

**Authors:** Shumin Zheng, Shaoqing Wang

**Affiliations:** 1School of Materials Science and Engineering, University of Science and Technology of China, Hefei 110016, China; 2Shenyang National Laboratory for Materials Science, Institute of Metal Research, Chinese Academy of Sciences, Shenyang 110016, China

**Keywords:** high-entropy alloys, first-principles calculation, maximum entropy, elastic property

## Abstract

The elastic properties of seventy different compositions were calculated to optimize the composition of a V–Mo–Nb–Ta–W system. A new model called maximum entropy approach (MaxEnt) was adopted. The influence of each element was discussed. Molybdenum (Mo) and tungsten (W) are key elements for the maintenance of elastic properties. The V–Mo–Nb–Ta–W system has relatively high values of *C*_44_, bulk modulus (*B*), shear modulus (*G*), and Young’s modulus (*E*), with high concentrations of Mo + W. Element W is brittle and has high density. Thus, low-density Mo can substitute part of W. Vanadium (V) has low density and plays an important role in decreasing the brittleness of the V–Mo–Nb–Ta–W system. Niobium (Nb) and tantalum (Ta) have relatively small influence on elastic properties. Furthermore, the calculated results can be used as a general guidance for the selection of a V–Mo–Nb–Ta–W system.

## 1. Introduction

In recent years, high entropy alloys (HEAs) have emerged as an interesting area of research [1]. HEAs have superior properties compared to conventional alloys [2]. Refractory high entropy alloys (RHEAs) were developed for high temperature use. RHEAs are mainly composed of Ti, V, Zr, Nb, Mo, Cr, Ta, W, and Hf. According to the literature, most RHEAs exceed the high use temperature of currently used refractory alloys Haynes^@^230^@^, MAR-M247^@^, INCONEL^@^718 [3], and conventional Ni-based superalloys [4]. This property makes RHEAs a promising candidate for the next generation of high-temperature applications. The VMoNbTaW alloy has received the most attention because of its characteristics, such as its good strength under extreme high temperature, but it is brittle between room temperature and 600 °C [5]. In a VMoNbTaW system, element W is a brittle element and has high density. Density is an important factor for transportation, especially for aircraft and aerospace. A high-temperature resistance is needed for turbine disks and blades, because the efficiency of gas turbines increases with working temperature [6].

Recently, many reports have shown that the best properties of RHEAs may generally be displaced from equilibrium compositions; thus, the studied compositions become complicated [7,8]. Some research focuses on the influence of elements on alloy properties, but most studies are often carried out for elements such as Al [9], Ti [10], Mo [11], and V [12]. It is feasible to study the influence of a single element, though single element optimization fails to meet application requirements most of the time. HEAs must have at least four elements in order to exhibit a high entropy effect [13]. Studying the influence of more than one element can enormously increase experimental efforts. First-principles calculation is an effective method for developing new RHEAs. Most data obtained in previous studies for RHEAs provide information for the hardness and compression of elements [3], but little is known about their elastic properties. Sufficiently large, homogeneous, and defect-free crystals are required to measure experimental elastic constants, so information on elastic properties is only available for a small portion of materials. Special quasi-random structure (SQS) [14] and coherent potential approximation adopted exact muffin-tin orbital (EMTO-CPA) are often used to predict the elastic properties of HEAs [15]. Elasticity is one of the fundamental properties to screen alloys and it directly relates to mechanical properties.

The present study reports a first-principles design of a VMoNbTaW alloy. The aims are to decrease the brittleness and density of a V–Mo–Nb–Ta–W system. The elastic properties of seventy different compositions were calculated. The influence of each element was discussed.

## 2. Methodology

CP2K was introduced for first-principles calculation and it is efficient for larger systems. CP2K is a quantum chemistry and solid-state physics software package [16]. QUICKSTEP was introduced to deal with the electronic structure. The Gaussian and plane wave (GPW) was used for the calculation of forces and energies [14]. Single-zeta valence Gaussian (SZV-MOLOPT-SR-GTH) was used as the basis set, while a 500Ry plane wave cutoff was used for the auxiliary grid. Fermi–Dirac smearing was used to accelerate the convergence to self-consistency with an electronic temperature of 300 K. In each self-consistent field (SCF) iteration step, the diagonalized Kohn–Sham matrix was introduced for solving eigenvalue issues. Additionally, Broyden mixing was used to accelerate the convergence to the total energy threshold. The value of the total energy threshold is 10^−7^ Hartree. A Broyden–Fletcher–Goldfarb–Shanno (BFGS) minimization algorithm was introduced to deal with the geometry optimization problems. The convergence criteria for the maximum geometry change and force were 1 × 10^−3^ Bohr and 1 × 10^−3^ Hartree/Bohr, respectively.

## 3. Maximum Entropy (MaxEnt) Model

MaxEnt structures were generated by a Monte Carlo simulation code in python. A repeat loop was written in the code to make sure all the elements were distributed homogeneously in the model [17]. In order to obtain a relatively homogeneous MaxEnt model, hundreds of structures were generated for selection. The screen criterion is the shortest distance between the same elements should locate in a narrow range—the narrower the better [18]. The most important advantage of the MaxEnt model is that it can demonstrate lattice distortion after relaxation. MaxEnt is a supercell model, while a 4 × 4 × 4 face-centered cubic (FCC) model contains 256 atoms and a 4 × 4 × 4 body-centered cubic (BCC) contains 128 atoms, so the MaxEnt model can present HEAs with complicated element concentrations. In order to test the accuracy and consistency of the MaxEnt model, ten MaxEnt models of BCC (TiZrNbMoV) were generated. Bulk moduli *B* and *C*_44_ were also calculated. All bulk moduli fluctuated around 143.3 (±2) GPa and all *C*_44_ fluctuated around 36.2 (±3) GPa. The scattered diagram is shown in Figure 1. Thus, the MaxEnt approach demonstrates a good consistence for each model. The MaxEnt approach has been elaborated in Reference [16]. The elastic properties of TaNbHfZrTi and CoCrFeNiMn were predicted based on the MaxEnt approach [18,19]. The accuracy of the predicted data was proven by experimental results [20,21]. Thus, the MaxEnt approach is accurate, believable, and suitable for the study of HEAs. 

All components of the VMoNbTaW alloy have a BCC lattice and, thus, the formation of BCC substitution solutions was the most probable. This was confirmed by diffraction analysis of these alloys [3]. The 4 × 4 × 4 MaxEnt model of V_0.1_Mo_0.2_Nb_0.1_Ta_0.4_W_0.2_ is shown as an example in Figure 2.

## 4. Elastic Properties

The calculated bulk modulus *B* and equilibrium lattice configuration were determined from the minima of the curves according to the Birch–Murnaghan equation of state (B–M EOS), as presented in Equation (1). *V*, *V*_0_, *B*, *E*, and *E*_0_ are volume, equilibrium volume, bulk modulus, total energy, and equilibrium energy, respectively. In order not to exceed the elastic limit, the changes in *V* should be kept within 3%.
(1)$$E\left(V\right)=E_{0}+\frac{9V_{0}B}{16}{\{\left[{\left(\frac{V_{0}}{V}\right)}^{\frac{2}{3}}-1\right]}^{3}B^{\prime }+{\left[{\left(\frac{V_{0}}{V}\right)}^{\frac{2}{3}}-1\right]}^{2}\left[6-4{\left(\frac{V_{0}}{V}\right)}^{\frac{2}{3}}\right]\}$$

The cubic crystal has three independent elastic constants: *C*_11_, *C*_12_, and *C*_44_. They can be calculated by applying small strains to the equilibrium lattice configuration, which transforms the lattice vector **a** according to the rule [22] shown in Equations (2) and (3).
a′ = a·(I + ε)(2)
(3)$$\varepsilon =\left(\begin{array}{ccc} e_{1} & e_{6}/2 & e_{5}/2\\ e_{6}/2 & e_{2} & e_{4}/2\\ e_{5}/2 & e_{4}/2 & e_{3} \end{array}\right)$$

***e*** = (*e*_1_, *e*_2_, *e*_3_, *e*_4_, *e*_5_, *e*_6_) is the strain vector. The different values of *e* were applied to the equilibrium lattice configuration according to Table 1. The value of σ should keep within the range of (−0.03, 0.03).

The following equations were used to calculate Shear modulus *G*, Young’s modulus *E*, and Poisson’s ratio *υ*.
(4)$$G=\frac{3C_{44}+C_{11}-C_{12}}{5}$$
(5)$$E=\frac{9BG}{3B+G}$$
(6)$$\nu =\frac{3B-2G}{2\left(3B+G\right)}$$

## 5. Results and Discussion

Due to the lack of experimental data of VMoNbTaW, the elastic properties of pure V, Mo, Nb, Ta, and W were calculated to prove the accuracy of the calculated data. Table 2 shows the elastic constants and moduli of V, Mo, Nb, Ta, and W. A comparison of calculated elastic properties with experimental data was made, and the agreement was found to be quite good. The accuracy of the calculated data in the present work was also proven by comparing with the calculated results in other studies.

Seventy different compositions of the V–Mo–Nb–Ta–W system were calculated. The results are shown in Table 3. W is brittle and has high density, so three concentrations (0.1, 0.2, and 0.3) of W were studied, while four concentrations each of V, Mo, Nb, and Ta (0.1, 0.2, 0.3, and 0.4) were studied.

All the structures were found to fulfill the mechanical stability criteria. The mechanical stability criterion of the cubic structure is *C*_11_ + 2*C*_12_ > 0, *C*_11_ > *C*_12_, *C*_44_ > 0. *C*_44_, *B*, *G*, *E*, *B*/*G* and *ν* are presented in scatter-plots in Figure 3, Figure 4, Figure 5 and Figure 6, respectively. The correspondence between the numbers in Table 3 and the X-axis is shown in Table 4.

### 5.1. C_44_

According to Reference [26], there is a monotonous relation between hardness and *C*_44_. In Figure 3, there is a regular distribution of all the points. They are distributed in two areas. The data points have the concentration of W + Mo ≥ 0.4 distributed at the top area. It is obvious that the values of *C*_44_ are bigger than the area below. There is also a data blank area between them. *C*_44_ increases with the increase of the W + Mo concentration. Thus, W and Mo show significant influence on *C*_44_. This may be due to the fact that the *C*_44_ of Mo (125 Gpa) and W (163 Gpa) are higher than the *C*_44_ of V (46 Gpa), Nb (31 Gpa), and Ta (82 Gpa). The densities of W and Mo are 19.350 g/cm^3^ and 10.390 g/cm^3^. In order to decrease the density and keep the high hardness of the V–Mo–Nb–Ta–W system, increasing Mo concentration and decreasing W concentration may be a feasible method.

### 5.2. Bulk Modulus

According to Reference [27], bulk modulus *B* can be used to describe the average atomic bond strength. The overall trend of the influence of alloying elements on *B* is shown in Figure 4. Figure 4a indicates that with the increase of V concentration, *B* decreases, while Figure 4b indicates *B* increases with the increase of Mo concentration. Additionally, Figure 4c indicates *B* decreases slightly with the increase of Nb concentration. Figure 4d shows that the concentration of Ta has no obvious influence on *B*. *B* increases with the increase of W concentration, as shown in Figure 4e. Furthermore, data points ran periodically with the changes of element concentrations. The trend in each period is the same as the overall trend of each element. For example, in Figure 4c, *B* decreases in the Nb = 0.1 area. This can be attributed to the increase in the concentration of element V. Arrow a in Figure 4e shows *B* decreases with the increase of Nb. Arrow c in Figure 4a shows *B* decreases with the decrease of W. A sharp variation in some points (1, 2, and 3) can be seen in Figure 4. For example, point 1 in Figure 4b shows that the initial concentration of W is 0.1, while the final concentration of W is 0.3. Thus, *B* increases sharply. In summary, Mo and W can help to increase *B*. Elements V and Nb have a negative effect on *B*. *B* has a high value in each period with W + Mo ≥ 0.4.

### 5.3. G, E, B/G, ν

The hardness of materials can be related to Young’s modulus *E* and the shear modulus *G* [28]. The general trend is that the larger these two moduli are, the harder the material. According to the Pugh criteria [29], materials with *B*/*G* < 2 are associated with brittleness; otherwise, the materials may behave as ductile. Materials with *υ* > 0.31 have good ductility. Otherwise, the materials are considered brittle.

Figure 5 shows *G* and *E* have the same trend, while *B*/*G* and *ν* also have the same trend. It can also be seen that there is an inverse relationship between them. Element V has a negative effect on *G* and *E*, and a positive effect on *B*/*G* and *ν*. Thus, the trend of *E*, *B*/*G*, and *ν* can be predicted from the trend of *G*. Figure 6 shows the trends of Mo, Nb, Ta, and W. It is obvious that W has a positive effect on *G* and *E* and exhibits a negative effect on *B*/*G* and *ν*, while elements Nb and Ta have no obvious effect. In summary, element V can help to increase the ductility of the V–Mo–Nb–Ta–W system. *G* and *E* have a relatively high value with W + Mo ≥ 0.4.

## 6. Conclusions

In order to improve the ductility and decrease the density of the V–Mo–Nb–Ta–W system, the elastic properties of seventy different compositions were studied. This work concludes as follows:
Mo and W are key elements in the V–Mo–Nb–Ta–W system. The V–Mo–Nb–Ta–W system has relatively high values of *C*_44_, *B*, *E*, and *G*, with high concentrations of Mo + W. The concentration of Mo + W shows the most prominent effect on *C*_44_. Due to the high density of W, element Mo can be used to substitute part of W. In this case, the concentration of W should be reduced.V has low density (6.110 g/cm^3^) and plays an important in decreasing the brittleness of the V–Mo–Nb–Ta–W system.In comparison, Nb and Ta have relatively small influence on elastic properties.

## Figures and Tables

**Figure 1 entropy-20-00965-f001:**
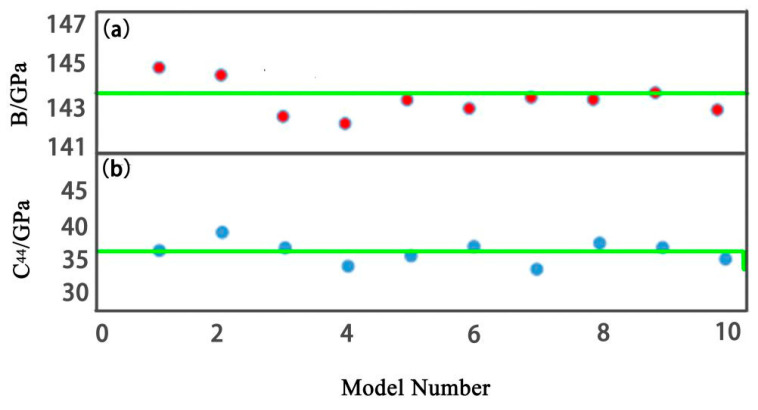
Bulk modulus *B* and *C*_44_ of ten maximum entropy (MaxEnt) models of a 4 × 4 × 4 BCC TiZrNbMoV alloy. (**a**) *B*, (**b**) *C*_44_.

**Figure 2 entropy-20-00965-f002:**
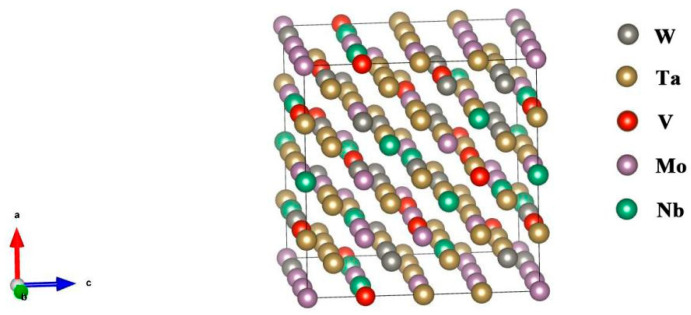
MaxEnt model of V_0.1_Mo_0.2_Nb_0.1_Ta_0.4_W_0.2_.

**Figure 3 entropy-20-00965-f003:**
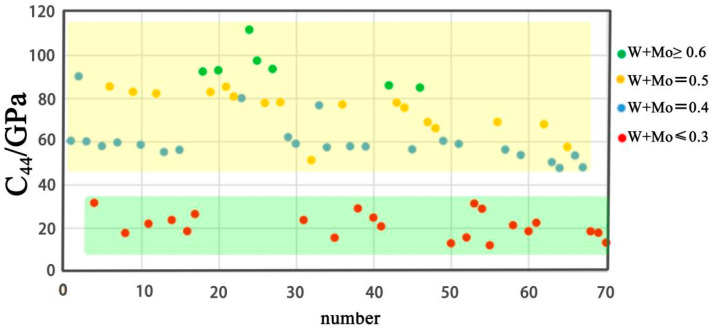
Scatter-plots of *C*_44_ of all seventy compositions.

**Figure 4 entropy-20-00965-f004:**
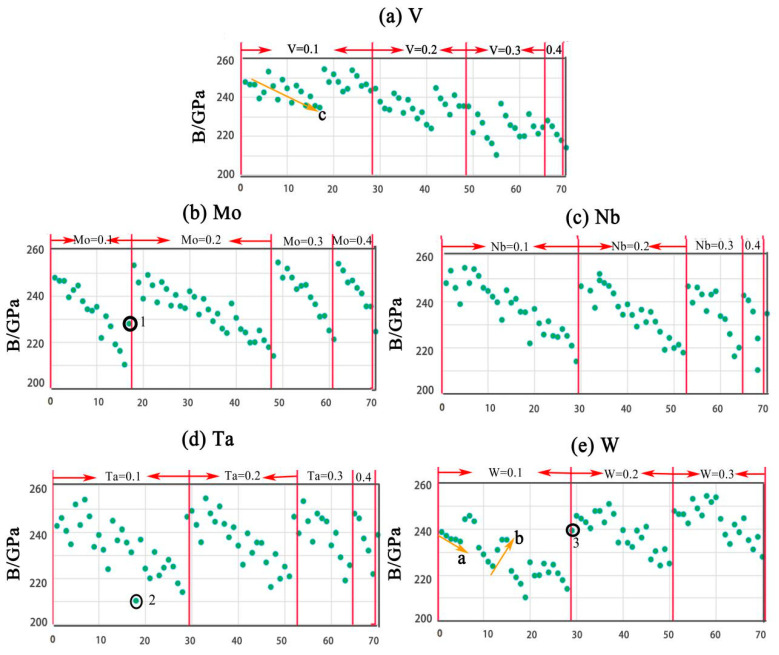
The trend of *B* along with element concentrations: (**a**) V, (**b**) Mo, (**c**) Nb, (**d**) Ta, and (**e**) W.

**Figure 5 entropy-20-00965-f005:**
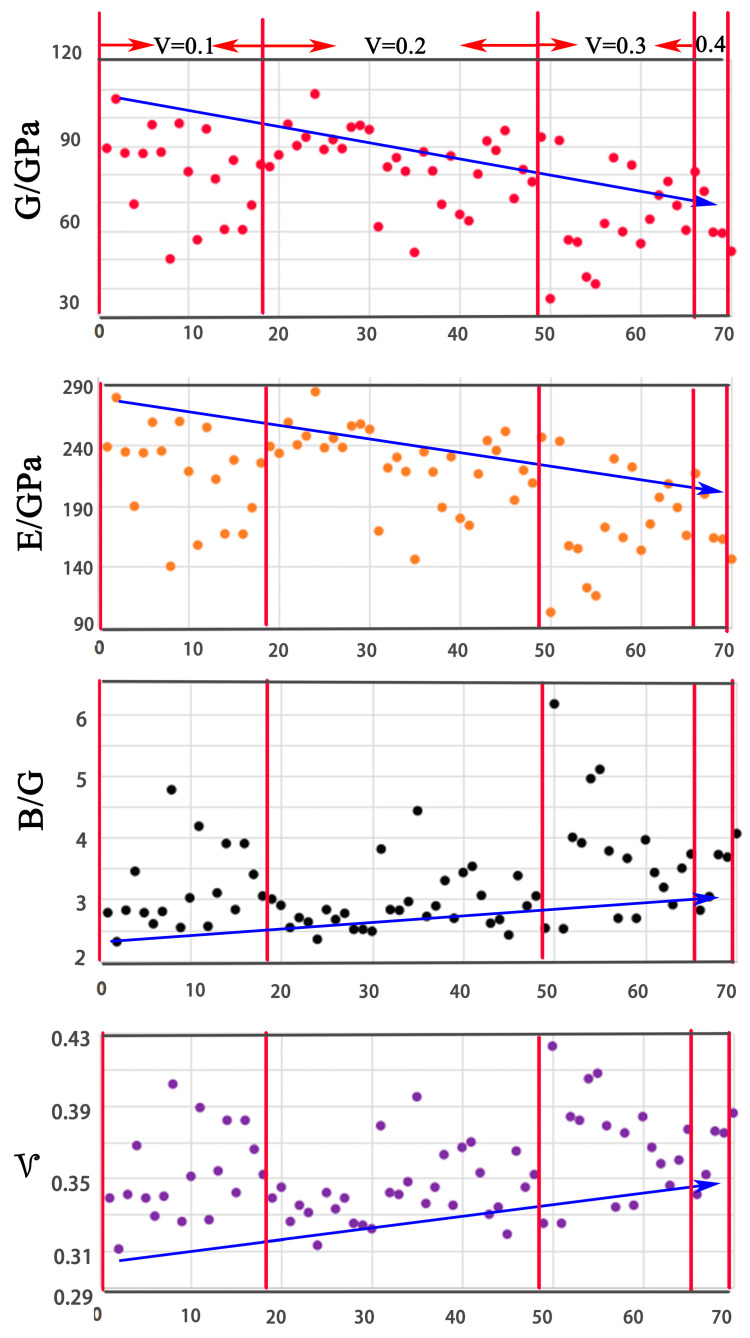
The trends of *G*, *E*, *B*/*G*, and *ν* along with V concentrations.

**Figure 6 entropy-20-00965-f006:**
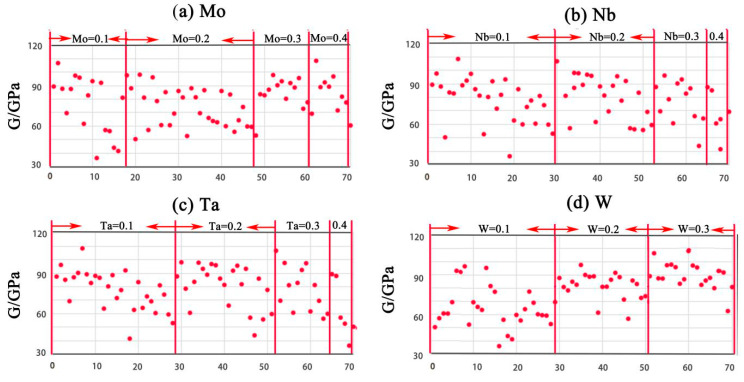
The trend of *G* along with element concentrations: (**a**) Mo, (**b**) Nb, (**c**) Ta, and (**d**) W.

**Table 1 entropy-20-00965-t001:** Vector strain and the corresponding energy.

Strain Vector *e*	The Corresponding Energy for Distorted Structure
(σ,0,0,0,0,0)	$E\left(\sigma \right)=E\left(0\right)+\frac{1}{2}C_{11}V_{0}\sigma^{2}$
(σ,σ,0,0,0,0)	$E\left(\sigma \right)=E\left(0\right)+\left(C_{11}+C_{12}\right)V_{0}\sigma^{2}$
(0,0,0,σ,σ,σ)	$E\left(\sigma \right)=E\left(0\right)+\frac{3}{2}C_{44}V_{0}\sigma^{2}$

**Table 2 entropy-20-00965-t002:** Elastic constants and moduli of V, Mo, Nb, Ta, and W.

Element		*C*_11_ (GPa)	*C*_12_ (GPa)	*C*_44_ (GPa)	*B* (GPa)	*E* (GPa)	*G* (GPa)	*ν*	ρ (g/cm^3^)	Tm (°C)
V	expt.	232.0 ^c^	119.0 ^c^	46.0 ^c^	155 ^b^	127.6 ^a^	46.7 ^a^	0.365 ^a^	6.11	1890
	present	261.0	109.5	45.5	159.7	154.0	57.6	0.340		
	other	205.0 ^c^	130.0 ^c^	5.0 ^c^	194.0 ^b^					
Mo	expt.	460.0 ^a^/450.0 ^c^	176.0 ^a^/173.0 ^c^	110.0 ^a^/125.0 ^c^	261.0 ^b^				10.39	2622
	present	407.8	150.0	135.5	236.0	335.3	132.7	0.263		
	other	406.0 ^c^	150.0 ^c^	107.0 ^c^	275.0 ^b^					
Nb	expt.	242 ^b^/253.0 ^c^	129 ^b^/133.0 ^c^	31.0 ^c^	169.0 ^b^				8.57	2468
	present	270.0	135.0	39.4	180.0	138.9	50.64	0.370		
	other	267.0 ^c^	147.0 ^c^	27.0 ^c^	171.0 ^b^					
Ta	expt.	267.0 ^a^/266.0 ^c^	161.0 ^a^/158.0 ^c^	82.5 ^a^/87.0 ^c^	191.0 ^b^	185.7 ^a^	69.2 ^a^	0.342 ^a^	16.65	2996
	present	287.0	151.0	72.7	195.6	189.6	70.82	0.338		
	other	291.0 ^c^	162.0 ^c^	84.0 ^c^	183.0 ^b^					
W	expt.	501.0 ^a^/533.0 ^c^	198.0 ^a^/205.0 ^c^	151.4 ^a^/163.0 ^c^	308.0 ^b^				19.35	3410
	present	558.0	181.5	173.8	307.9	451.0	179.6	0.256		
	other	553.0 ^c^	207.0 ^c^	178.0 ^c^	306.0 ^b^					

^a^ Reference [23]. ^b^ Reference [24]. ^c^ Reference [25].

**Table 3 entropy-20-00965-t003:** Elastic constants and moduli of RHEAs. *B* (GPa), *E* (GPa), *G* (GPa) and *υ* represent the bulk modulus, Young’s modulus, shear modulus, and Poisson’s ratio and *a* (Ả) stands for equilibrium lattice constants.

System	*C* _11_	*C* _12_	*C* _44_	*B*	*G*	*E*	*B*/*G*	*ν*	*a*
(1)V_0.1_Mo_0.1_Nb_0.1_Ta_0.4_W_0.3_	424.5	159.4	60.1	247.7	89.0	238.6	2.78	0.339	3.199
(2)V_0.1_Mo_0.1_Nb_0.2_Ta_0.3_W_0.3_	421.8	158.7	89.8	246.4	106.5	279.2	2.31	0.311	3.200
(3)V_0.1_Mo_0.1_Nb_0.3_Ta_0.2_W_0.3_	418.1	160.5	59.8	246.3	87.4	234.5	2.82	0.341	3.200
(4)V_0.1_Mo_0.1_Nb_0.3_Ta_0.3_W_0.2_	407.4	155.2	31.5	239.3	69.3	189.5	3.45	0.368	3.212
(5)V_0.1_Mo_0.1_Nb_0.4_Ta_0.1_W_0.3_	417.9	154.6	57.6	242.4	87.2	233.7	2.78	0.339	3.202
(6)V_0.1_Mo_0.2_Nb_0.1_Ta_0.3_W_0.3_	407.0	176.1	85.1	253.1	97.4	259.0	2.60	0.329	3.187
(7)V_0.1_Mo_0.2_Nb_0.1_Ta_0.4_W_0.2_	419.5	158.8	59.3	245.7	87.7	235.2	2.80	0.340	3.199
(8)V_0.1_Mo_0.2_Nb_0.1_Ta_0.5_W_0.1_	370.2	172.9	17.6	238.6	50.0	140.2	4.77	0.402	3.212
(9)V_0.1_Mo_0.2_Nb_0.2_Ta_0.2_W_0.3_	409.8	168.5	82.6	248.9	97.8	259.5	2.54	0.326	3.190
(10)V_0.1_Mo_0.2_Nb_0.2_Ta_0.3_W_0.2_	397.4	168.5	58.2	244.5	80.8	218.4	3.02	0.351	3.200
(11)V_0.1_Mo_0.2_Nb_0.2_Ta_0.4_W_0.1_	382.5	164.3	21.9	237.1	56.7	157.7	4.18	0.389	3.212
(12)V_0.1_Mo_0.2_Nb_0.3_Ta_0.1_W_0.3_	401.7	167.9	81.9	245.8	95.9	254.7	2.56	0.327	3.190
(13)V_0.1_Mo_0.2_Nb_0.3_Ta_0.2_W_0.2_	393.9	167.3	54.9	242.9	78.2	212.0	3.10	0.354	3.201
(14)V_0.1_Mo_0.2_Nb_0.3_Ta_0.3_W_0.1_	389.9	158.5	23.6	235.6	60.4	167.0	3.90	0.382	3.212
(15)V_0.1_Mo_0.2_Nb_0.4_Ta_0.2_W_0.2_	411.4	154.7	55.9	240.3	84.8	227.8	2.83	0.342	3.201
(16)V_0.1_Mo_0.2_Nb_0.4_Ta_0.2_W_0.1_	399.9	153.5	18.4	235.4	60.3	166.8	3.90	0.382	3.214
(17)V_0.1_Mo_0.2_Nb_0.5_Ta_0.1_W_0.1_	411.9	145.9	26.3	234.5	69.0	188.5	3.40	0.366	3.214
(18)V_0.1_Mo_0.3_Nb_0.1_Ta_0.2_W_0.3_	348.1	207.5	92.0	254.3	83.3	225.4	3.05	0.352	3.176
(19)V_0.1_Mo_0.3_Nb_0.1_Ta_0.3_W_0.2_	380.1	181.6	82.5	247.7	82.5	239.0	3.00	0.339	3.188
(20)V_0.1_Mo_0.3_Nb_0.2_Ta_0.1_W_0.3_	355.6	199.7	92.6	251.7	86.7	233.4	2.90	0.345	3.177
(21)V_0.1_Mo_0.3_Nb_0.2_Ta_0.2_W_0.2_	403.1	170.1	85.0	247.8	97.6	258.8	2.54	0326	3.189
(22)V_0.1_Mo_0.3_Nb_0.3_Ta_0.1_W_0.2_	381.8	173.3	80.5	242.8	90.0	240.3	2.70	0.335	3.190
(23)V_0.1_Mo_0.3_Nb_0.3_Ta_0.2_W_0.1_	394.7	169.0	79.7	244.2	93.0	247.6	2.63	0.331	3.190
(24)V_0.1_Mo_0.4_Nb_0.1_Ta_0.1_W_0.3_	391.7	184.8	111.3	253.8	108.2	284.1	2.35	0.313	3.167
(25)V_0.1_Mo_0.4_Nb_0.1_Ta_0.2_W_0.2_	352.2	200.2	97.0	250.9	88.6	237.9	2.83	0.342	3.176
(26)V_0.1_Mo_0.4_Nb_0.1_Ta_0.3_W_0.1_	397.8	169.6	77.5	245.7	92.1	245.7	2.67	0.333	3.188
(27)V_0.1_Mo_0.4_Nb_0.2_Ta_0.1_W_0.2_	356.7	191.4	93.2	246.5	88.9	238.2	2.77	0.339	3.178
(28)V_0.1_Mo_0.4_Nb_0.2_Ta_0.2_W_0.1_	409.4	160.2	77.8	243.3	96.5	255.8	2.51	0.325	3.189
(29)V_0.2_Mo_0.1_Nb_0.1_Ta_0.3_W_0.3_	444.7	144.2	61.8	244.3	97.1	257.4	2.51	0.324	3.174
(30)V_0.2_Mo_0.1_Nb_0.2_Ta_0.2_W_0.3_	439.0	136.8	58.7	237.6	95.	253.1	2.48	0.322	3.177
(31)V_0.2_Mo_0.1_Nb_0.2_Ta_0.3_W_0.2_	391.6	155.3	23.5	234.1	61.3	169.3	3.81	0.379	3.188
(32)V_0.2_Mo_0.1_Nb_0.3_Ta_0.1_W_0.3_	406.0	147.2	51.2	233.4	82.4	221.4	2.83	0.342	3.179
(33)V_0.2_Mo_0.1_Nb_0.2_Ta_0.2_W_0.3_	374.8	175.4	76.4	241.9	85.7	230.1	2.82	0.341	3.165
(34)V_0.2_Mo_0.2_Nb_0.1_Ta_0.3_W_0.2_	395.2	161.6	57.1	239.5	80.9	218.3	2.96	0.348	3.175
(35)V_0.2_Mo_0.2_Nb_0.1_Ta_0.4_W_0.1_	375.4	160.9	15.4	231.8	52.3	145.9	4.43	0.395	3.188
(36)V_0.2_Mo_0.2_Nb_0.2_Ta_0.1_W_0.3_	377.5	169.1	76.8	238.6	87.8	234.6	2.72	0.336	3.167
(37)V_0.2_Mo_0.2_Nb_0.2_Ta_0.2_W_0.2_	392.7	160.0	57.6	234.0	81.1	218.0	2.89	0.345	3.189
(38)V_0.2_Mo_0.2_Nb_0.2_Ta_0.3_W_0.1_	402.0	142.4	28.9	228.9	69.2	188.8	3.30	0.363	3.189
(39)V_0.2_Mo_0.2_Nb_0.3_Ta_0.1_W_0.2_	405.2	145.7	57.4	232.2	86.3	230.5	2.69	0.335	3.177
(40)V_0.2_Mo_0.2_Nb_0.3_Ta_0.2_W_0.1_	395.1	141.0	24.7	225.7	65.7	179.7	3.43	0.367	3.190
(41)V_0.2_Mo_0.2_Nb_0.4_Ta_0.1_W_0.1_	394.2	138.2	20.6	223.8	63.4	173.9	3.53	0.370	3.191
(42)V_0.2_Mo_0.3_Nb_0.1_Ta_0.1_W_0.3_	340.0	196.9	85.6	244.6	79.9	216.3	3.06	0.353	3.153
(43)V_0.2_Mo_0.3_Nb_0.1_Ta_0.2_W_0.2_	389.4	164.2	77.7	239.3	91.6	243.8	2.61	0.330	3.164
(44)V_0.2_Mo_0.3_Nb_0.2_Ta_0.1_W_0.2_	380.0	164.3	75.3	236.2	88.3	235.7	2.67	0.334	3.165
(45)V_0.2_Mo_0.3_Nb_0.2_Ta_0.2_W_0.1_	436.4	128.1	56.1	230.9	95.3	251.4	2.42	0.319	3.177
(46)V_0.2_Mo_0.4_Nb_0.1_Ta_0.1_W_0.2_	309.4	206.7	84.6	241.0	71.3	194.7	3.38	0.365	3.152
(47)V_0.2_Mo_0.4_Nb_0.1_Ta_0.2_W_0.1_	369.7	168.1	68.7	235.3	81.5	219.3	2.89	0.345	3.164
(48)V_0.2_Mo_0.4_Nb_0.2_Ta_0.1_W_0.1_	361.0	172.5	65.9	235.3	77.2	208.8	3.05	0.352	3.165
(49)V_0.3_Mo_0.1_Nb_0.1_Ta_0.2_W_0.3_	425.1	140.1	60.0	235.1	93.0	246.6	2.53	0.325	3.152
(50)V_0.3_Mo_0.1_Nb_0.1_Ta_0.4_W_0.1_	315.8	174.6	12.8	221.7	35.9	102.4	6.16	0.423	3.177
(51)V_0.3_Mo_0.1_Nb_0.2_Ta_0.1_W_0.3_	419.9	136.7	58.6	231.1	91.8	243.3	2.52	0.325	3.153
(52)V_0.3_Mo_0.1_Nb_0.2_Ta_0.2_W_0.2_	384.7	147.7	15.5	226.7	56.7	157.1	4.00	0.384	3.164
(53)V_0.3_Mo_0.1_Nb_0.2_Ta_0.3_W_0.1_	343.1	156.7	31.2	218.9	56.0	154.8	3.91	0.382	3.177
(54)V_0.3_Mo_0.1_Nb_0.3_Ta_0.2_W_0.1_	303.9	172.1	28.8	216.0	43.6	122.6	4.95	0.405	3.178
(55)V_0.3_Mo_0.1_Nb_0.4_Ta_0.1_W_0.1_	323.5	153.4	11.9	210.1	41.1	115.9	5.10	0.408	3.180
(56)V_0.3_Mo_0.2_Nb_0.1_Ta_0.1_W_0.3_	307.4	201.1	68.8	236.6	62.5	172.4	3.78	0.379	3.139
(57)V_0.3_Mo_0.2_Nb_0.1_Ta_0.2_W_0.2_	404.2	143.4	56.0	230.3	85.7	228.8	2.69	0.334	3.155
(58)V_0.3_Mo_0.2_Nb_0.1_Ta_0.3_W_0.1_	374.1	140.7	21.2	225.5	59.6	164.0	3.66	0.375	3.164
(59)V_0.3_Mo_0.2_Nb_0.2_Ta_0.1_W_0.2_	394.2	138.9	53.5	224.0	83.1	222.0	2.69	0.335	3.156
(60)V_0.3_Mo_0.2_Nb_0.2_Ta_0.2_W_0.1_	367.6	145.6	18.4	219.6	55.4	153.4	3.96	0.384	3.169
(61)V_0.3_Mo_0.2_Nb_0.3_Ta_0.1_W_0.1_	388.6	135.4	22.3	219.8	64.0	175.0	3.43	0.367	3.165
(62)V_0.3_Mo_0.3_Nb_0.1_Ta_0.1_W_0.2_	337.5	178.0	67.7	231.2	72.5	197.0	3.19	0.358	3.141
(63)V_0.3_Mo_0.3_Nb_0.1_Ta_0.2_W_0.1_	382.2	146.2	50.2	224.9	77.3	208.2	2.91	0.346	3.151
(64)V_0.3_Mo_0.3_Nb_0.2_Ta_0.1_W_0.1_	375.9	174.2	47.6	221.1	68.9	188.0	3.50	0.360	3.156
(65)V_0.3_Mo_0.4_Nb_0.1_Ta_0.1_W_0.1_	310.4	181.4	57.2	224.4	60.1	165.7	3.73	0.377	3.140
(66)V_0.4_Mo_0.1_Nb_0.1_Ta_0.1_W_0.3_	390.5	146.4	53.3	227.8	80.7	216.7	2.82	0.341	3.127
(67)V_0.4_Mo_0.2_Nb_0.1_Ta_0.1_W_0.2_	375.5	149.6	47.8	224.9	73.9	199.8	3.04	0.352	3.128
(68)V_0.4_Mo_0.2_Nb_0.1_Ta_0.2_W_0.1_	382.1	139.9	18.3	220.6	59.4	163.6	3.72	0.376	3.140
(69)V_0.4_Mo_0.2_Nb_0.2_Ta_0.1_W_0.1_	378.7	136.1	17.7	217.7	59.1	162.6	3.68	0.375	3.141
(70)V_0.5_Mo_0.2_Nb_0.1_Ta_0.1_W_0.1_	363.3	139.1	13.1	213.8	52.7	146.1	4.06	0.386	3.115

**Table 4 entropy-20-00965-t004:** The correspondence between the numbers in Table 3 and the X-axis. The first point in the X-axis involved with V is the data of number 1 in Table 3. The first point in the X-axis involved with Ta is the data of number 5 in Table 3.

Table 2	V	Mo	Nb	Ta	W	Table 2	V	Mo	Nb	Ta	W	Table 2	V	Mo	Nb	Ta	W
1	1	1	1	65	51	25	25	63	8	36	37	49	49	10	18	45	67
2	2	2	30	52	52	26	26	64	9	58	7	50	50	11	19	69	16
3	3	3	53	29	53	27	27	65	36	8	38	51	51	12	46	17	68
4	4	4	54	53	29	28	28	66	37	37	8	52	52	13	47	46	46
5	5	5	65	1	54	29	29	6	10	59	61	53	53	14	48	63	17
6	6	18	2	54	55	30	30	7	38	38	62	54	54	15	63	47	18
7	7	19	3	66	30	31	31	8	39	60	39	55	55	16	69	18	19
8	8	20	4	70	1	32	32	9	60	9	63	56	56	39	20	19	69
9	9	21	34	30	56	33	33	30	11	39	64	57	57	40	21	48	47
10	10	22	32	55	31	34	34	31	12	61	40	58	58	41	22	64	20
11	11	23	33	67	2	35	35	32	13	68	9	59	59	42	49	20	48
12	12	24	55	2	57	36	36	33	40	10	65	60	60	43	50	49	21
13	13	25	56	31	32	37	37	34	41	40	41	61	61	44	64	21	22
14	14	26	57	56	3	38	38	35	42	62	10	62	62	59	23	22	49
15	15	27	66	3	33	39	39	36	61	11	42	63	63	60	24	50	23
16	16	28	67	32	4	40	40	37	62	41	11	64	64	61	51	23	24
17	17	29	70	4	5	41	41	38	68	12	12	65	65	70	25	24	25
18	18	49	5	33	58	42	42	55	14	13	66	66	66	17	26	25	70
19	19	50	6	57	34	43	43	56	15	42	43	67	67	45	27	26	50
20	20	51	34	5	59	44	44	57	43	14	44	68	68	46	28	51	26
21	21	52	35	34	35	45	45	58	44	43	13	69	69	47	52	27	27
22	22	53	58	6	36	46	46	67	16	15	45	70	70	48	29	28	28
23	23	54	59	35	6	47	47	68	17	44	14						
24	24	62	7	7	60	48	48	69	45	16	15

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
