# Peer review of "First-Principles Design of Refractory High Entropy Alloy VMoNbTaW"

_entropy, 2018, doi:10.3390/e20120965_

Round 1

Reviewer 1 Report

This DFT-based work studied 70 compositions of V-Mo-Nb-Ta-W for their elastic properties and summarized element-specific influences on each property. The calculations are properly set up and the conclusions are supported by the data. I recommend publication with minor revision.

- Table 2 is hard to read. The rows do not align well.

- The ranges of V in Eq.1 and delta in Table 1 need to be specified so that others can reproduce the results. Sometimes the elastic properties are sensitive to these settings.

- It may be better to plot Figure 3 as C44 vs. X(W)+X(Mo). It will convey more information. Besides, if in the context it is said that 0.3<W+Mo<0.4 is not studied, why does it say W+Mo<0.4 in Figure 3? Shouldn't it be <0.3? It the x-axis is W+Mo, such ambiguity can be avoided.

- It may be better to plot Figure 4 as contour plots, i.e., z=B, xy=(W, Mo), (V, Nb), (W+Mo, Ta)

Author Response

  We write to resubmit our manuscript having considered carefully the comments and suggestions, and having responded accordingly. It is our hope that you find the inputs and corrections as commensurate with your brilliant observations. Please see below the response to the reviewers.

1.Table 2 is hard to read. The rows do not align well.

Answer: As the referee’s comment pointed out. We revised Table 2.

2.The ranges of V in Eq.1 and delta in Table 1 need to be specified so that others can reproduce the results. Sometimes the elastic properties are sensitive to these settings.

Answer: We appreciate this comment. We revised the manuscript, please see Page 3, line 96, V, V0, B, E and E0 are volume, equilibrium volume, bulk modulus, total energy and equilibrium energy, respectively. In order not to exceed the elastic limit, the changes in V should keep within 3%. Please see Page 4, line 103,The value of σ should keep within the range of (-0.03,0.03).

3.It may be better to plot Figure 3 as C44 vs. X(W)+X(Mo). It will convey more information. Besides, if in the context it is said that 0.3<W+Mo<0.4 is not studied, why does it say W+Mo<0.4 in Figure 3? Shouldn't it be <0.3? It the x-axis is W+Mo, such ambiguity can be avoided.

Answer: We thank you for your observation. We revised Figure 3. please see Figure 3. We revised the paper to avoid ambiguity.

4. It may be better to plot Figure 4 as contour plots, i.e., z=B, xy=(W, Mo), (V, Nb), (W+Mo, Ta)

Answer: We thank you for your observation. We tried to revise Figure 4 as your suggestion. However, I can not find a way to better present our data. Would it be ok if I keep it as it is?

Finally, we appreciate very much for your time in editing our manuscript and the referees for their valuable suggestions and comments. I am looking forward to hearing from your final decision when it is made.

Yours sincerely,

Zheng Shumin

Reviewer 2 Report

This paper reports the first principles design of high entropy alloy (HEA) VMoNbTaW. They calculated the elastic properties of VMoNbTaW with seventy different compositions.  They have found two strategies to improve the ductility and decrease the density of V-Mo-Nb-Ta-W system. The first one is that Mo and W are key elements to obtain high value of C44, bulk modulus, shear modulus and Young’s modulus. The second one is that V plays an important role in deceasing the brittleness.

The paper is well organized and stimulates researchers in the field of HEA. So I believe the manuscript meets all criteria necessary for Entropy. But, before the acceptance, I recommend the authors to address the comment listed below.

(1)  I recommend the authors to add a brief introduction of elastic properties of typical materials including HEA. I think it is useful for the general readers.

Author Response

  We write to resubmit our manuscript having considered carefully the comments and suggestions, and having responded accordingly. It is our hope that you find the inputs and corrections as commensurate with your brilliant observations. Please see below the response to the reviewers.

(1)   I recommend the authors to add a brief introduction of elastic properties of typical materials including HEA. I think it is useful for the general readers.

Answer: We appreciate this comment. We revised the manuscript, please see Page 1, line 44, “Sufficiently large, homogeneous and defect-free crystals are required to measure experimental elastic constants, so elastic properties are only available for a small portion of materials. Special Quasi-random Structures (SQS) [14] and Coherent Potential Approximation adopted Exact Muffin-Tin Orbital (EMTO-CPA) are often used to predict elastic properties of HEAs [15].  

Finally, we appreciate very much for your time in editing our manuscript and the referees for their valuable suggestions and comments. I am looking forward to hearing from your final decision when it is made.

Yours sincerely,

Zheng Shumin

Reviewer 3 Report

In recent years, the approach to the creation of multicomponent alloys having as a matrix several elements in equal equiatomic proportions, has been quite interesting. These multicomponent alloys, called high-entropy alloys, can have a number of valuable properties, such as high strength and hardness, wear resistance, high temperature strength, corrosion resistance, which makes them attractive for research. In this regard, the submitted manuscript, aimed at the first-principles design of VMoNbTaW alloy, is certainly topical. 

The data obtained by the authors are of interest to researchers engaged in the development of new high-entropy alloys for high-temperature applications. Methods are well described. The illustrations and tables given in the manuscript fully disclose the results of the research. It is certainly worth of publication, but some minor revisions are suggested to improve its quality. More specifically: 

- Equation (1) is not deciphered. It is not clear what the letters in the equation mean. The author must give an unambiguous interpretation of each literal value in the equation. 

- Fig. 1 and Fig. 3 has a very low resolution. 

- Some minor mistakes have been detected in the text, is recommended to revise and correct some gramatical errors.

Author Response

  We write to resubmit our manuscript having considered carefully the comments and suggestions, and having responded accordingly. It is our hope that you find the inputs and corrections as commensurate with your brilliant observations. Please see below the response to the reviewers.

1. Equation (1) is not deciphered. It is not clear what the letters in the equation mean. The author must give an unambiguous interpretation of each literal value in the equation. 

Answer: We appreciate this comment. We revised the manuscript, please see Page 3 line 96, V, V0, B, E and E0 are volume, equilibrium volume, bulk modulus, total energy and equilibrium energy, respectively. In order not to exceed the elastic limit, the changes in V should keep in 3%.     

2. Fig. 1 and Fig. 3 has a very low resolution. 

Answer: We appreciate this comment. The resolution of this two figures have been set as 1000.

3. Some minor mistakes have been detected in the text, is recommended to revise and correct some gramatical errors.

Answer: Thankyou for your suggestion. I have asked a native English speaker to polish the paper.

Finally, we appreciate very much for your time in editing our manuscript and the referees for their valuable suggestions and comments. I am looking forward to hearing from your final decision when it is made.

Yours sincerely,

Zheng Shumin
